# Increased Pericardial Adipose Tissue in Smokers

**DOI:** 10.3390/jcm10153382

**Published:** 2021-07-30

**Authors:** Gregor S. Zimmermann, Tobias Ruether, Franz von Ziegler, Martin Greif, Janine Tittus, Jan Schenzle, Christoph Becker, Alexander Becker

**Affiliations:** 1Department of Internal Medicine I, School of Medicine & Klinikum rechts der Isar, Technical University of Munich, 81675 Munich, Germany; 2Department of Psychiatry, Ludwig-Maximilians-University Munich, 80336 Munich, Germany; tobias.ruether@med.uni-muenchen.de; 3Department of Cardiology, Ludwig-Maximilians-University Munich, 81377 Munich, Germany; von_ziegler@yahoo.com (F.v.Z.); greif_martin@gmx.de (M.G.); alexander.becker@med.uni-muenchen.de (A.B.); 4Department of Cardiology, Klinikum Augustinum Munich, 81375 Munich, Germany; tittus@med.augustinum.de; 5Department of Cardiology, Klinikum Garmisch-Partenkirchen, 82467 Garmisch-Partenkirchen, Germany; jan.schenzle@klinikum-gap.de; 6Department of Clinical Radiology, Ludwig-Maximilians-University Munich, 81377 Munich, Germany; chbecker@stanford.edu; 7Cardiovascular Imaging Division, Department of Radiology, Stanford University Medical Center, Stanford, CA 94305, USA

**Keywords:** pericardial adipose tissue, risk prediction, metabolic risk factors, smoking

## Abstract

Background: Pericardial adipose tissue (PAT), a visceral fat depot directly located to the heart, is associated with atherosclerotic and inflammatory processes. The extent of PAT is related to the prevalence of coronary heart disease and might be used for cardiovascular risk prediction. This study aimed to determine the effect of smoking on the extent of PAT. Methods: We retrospectively examined 1217 asymptomatic patients (490 females, age 58.3 ± 8.3 years, smoker *n* = 573, non-smoker *n* = 644) with a multislice CT scanner and determined the PAT volume. Coronary risk factors were determined at inclusion, and a multivariate analysis was performed to evaluate the influence of smoking on PAT independent from accompanying risk factors. Results: The mean PAT volume was 215 ± 107 mL in all patients. The PAT volume in smokers was significantly higher compared to PAT volume in non-smokers (231 ± 104 mL vs. 201 ± 99 mL, *p* = 0.03). Patients without cardiovascular risk factors showed a significantly lower PAT volume (153 ± 155 mL, *p* < 0.05) compared to patients with more than 1 risk factor. Odds ratio was 2.92 [2.31, 3.61; *p* < 0.001] for elevated PAT in smokers. Conclusion: PAT as an individual marker of atherosclerotic activity and inflammatory burden was elevated in smokers. The finding was independent from metabolic risk factors and might therefore illustrate the increased inflammatory activity in smokers in comparison to non-smokers.

## 1. Introduction

Coronary artery disease (CAD) is one of the leading causes of death in industrialized countries [1]. Many non-invasive strategies have been developed to identify the individual risk for coronary artery disease in the last decades. The determination of coronary atherosclerosis by cardiac computed tomography (CT) is an established method for the evaluation of cardiovascular risk and provides a more accurate prediction for the individual patients than conventional risk factors scores [2,3,4,5].

Pericardial adipose tissue (PAT), a visceral fat depot surrounding the heart, is a further marker of atherosclerotic processes, which can be determined by non-contrast cardiac CT [6]. There is a significant association between PAT volume, prevalence of coronary heart disease, and risk for future cardiac events [7,8,9]. PAT is involved in multiple steps of atherosclerosis [7,10,11]. Especially, PAT increases pro-inflammatory stimuli and leads to oxidative stress by secreting multiple endocrine (e.g., leptin, resistin) and paracrine factors (e.g., tumor necrosis factor alpha (TNF-α), monocyte chemoattractant protein-1 (MCP-1), intercellular adhesion molecule 1 (ICAM-1) and interleukins (IL1, IL1β, IL-1Ra, IL-6, IL-8) in patients with CAD) [12,13]. Due to its proximity to the coronary arteries PAT influences cardiac structure and myocardial microcirculation [14,15]. An increase in PAT volume is therefore linked to an accelerated inflammation [12,15,16,17,18].

PAT could already show its predictive value for future cardiac events in addition to coronary calcium scoring. To quantify the extent of PAT, the volume of PAT can be determined reliable in a native CT scan, e.g., in a scan for coronary calcium [7,19,20].

Beside the metabolic risk factors, diabetes mellitus and hyperlipidemia, smoking is a major risk factor for CAD, particularly because of its impact on systematic inflammation [21,22,23]. These processes result in endothelial dysfunction, atherosclerosis, and cardiovascular events [21,22,24]. In smokers, significantly higher values of phosholipase A2, high-sensitive CRP (hs-CRP) and other inflammatory markers can be found [22,24,25,26]. These specific pathogenic effects of smoking on vascular function differ from the effects of other cardiovascular risk factors and may be a reason why screening of coronary calcifications alone underestimates the effect of smoking on the coronary arteries [4,27]. Smokers showed a significantly higher risk compared to non-smokers with equal calcium scores [4]. To assess the effect of smoking on the inflammatory active structures, we determined the volume of PAT in a large cohort of smokers and non-smokers.

## 2. Materials and Methods

### 2.1. Study Protocol

In this retrospective single center study, we examined 1217 consecutive patients referred to the outpatient department of our hospital by a primary care physician for preventive cardiological examination between 2008 and 2015.

Beside clinical examination, ECG, stress ECG, and echocardiography all patients underwent coronary calcium screening by computed tomography [4,19,28]. These images were used for the determination of pericardial fat as described earlier [4,7,19]. This study has been performed in accordance with the ethical standards laid down in the 1964 Declaration of Helsinki and its later amendment.

### 2.2. Risk Factors

We evaluated conventional cardiovascular risk factors for all patients by means of a personal interview and by reviewing their medical records. In addition, arterial blood pressure, lipid status (LDL cholesterol level, HDL cholesterol level, triglyceride level) and blood glucose level were determined in the fasting state following a standardized protocol, as described in earlier studies of our group [4,10,19]. Hyperlipidemia was also determined by statine use. Smoker and non-smoker were defined according the Centers for Disease Control and Prevention (CDC) definition. Smoking status was positive if the patient is a current smoker. Non-smoker was defined by CDC as an adult who has never smoked, or who has smoked less than 100 cigarettes.

### 2.3. Pericardial Fat Assessment Protocol

Pericardial adipose tissue measurements were performed using images acquired for coronary calcium (CAC) screening following a standardized protocol [10,19]. Pericardial fat was defined as epicardial fat (adipose tissue within the pericardium) plus paracardial fat (adipose tissue on the external surface of the parietal pericardium) as depicted in Figure 1. During one end-inspiratory breath-holding period ECG-triggered scans of 100-ms duration were acquired at 80% of the R–R interval. A total of 40 slices, 0.3-mm slice thickness, were obtained covering the whole heart as described earlier [4,10]. CAC scanning was performed using a Siemens multislice CT scanner (Somatom Sensation 16 or 64-slice, Siemens Medical Solutions, Forchheim, Germany) in the high-resolution mode, as described earlier by our group [4,10].

The volume of pericardial adipose tissue was measured in millilitre (mL) using the volume analysis software of our cardiac workstation (Leonardo; Siemens Medical Solutions, Forchheim, Germany). As described earlier the superior cut-off point in the axial slices was the bifurcation of the pulmonary artery. Inferiorly, the volume analyzed was segmented up to the intraabdominal adipose tissue. The anterior border was defined by the anterior chest wall. The posterior border was defined by the esophagus and the descending aorta [7,10,19]. The borders were set manually by the investigator.

To isolate the adipose tissue (fat)-containing voxels, a threshold of −250 to −30 HU was applied after the segmentation of the heart and surrounding adipose tissue from the remainder of the thorax (see Figure 1). These voxels were then summed to obtain adipose tissue volume (mL) [7,10,19]. In order to assess the accuracy of PAT determination, PAT measurements were performed by two independent investigators who were blinded to patients’ clinical records.

### 2.4. Statistical Analysis

Statistical analyses were performed using the SPSS software package (version 18.0, SPSS Inc. Chicago, IL, USA). All values are expressed as mean score ± standard deviation (SD), exceptions were indicated. Because of the non-normality, statistical analysis was performed on the base 10 log of the transformed PAT volume. Continuous variables compared between groups were assessed using the Kruskal–Wallis test in case of more than two groups; in case of two groups we used the Mann–Whitney *U* test.

To assess the relationship between risk factors and PAT, we performed an univariate Cox regression analysis to calculate odds ratio and 95% confidence interval of PAT in dependence of cardiovascular risk factors. Patients without cardiovascular risk factors served as the reference group. A multivariate regression model was built, where potential confounders were chosen from a set of candidate confounders using a backward elimination algorithm at the significance level of 5%. To account for the inflation of the type I error due to multiple testing, we performed the Bonferroni adjustment. The significance level was set at 0.05/4 = 0.0125. To assess multicollinearity in our population, we calculated variance inflation factor (VIF) for the different cardiovascular risk factors.

## 3. Results

### 3.1. Study Population

In all 1217 individuals (490 female and 727 male, age 58.3 ± 8.3 years), PAT volume could be determined. The distribution of cardiovascular risk factors in smokers and non-smokers is shown in Table 1. The mean number of risk factors was 1.4 in non-smokers and 2.5 in smokers. Apart from smoking, there was no difference in age, sex, and risk factor distribution between smokers and non-smokers. Baseline characteristics of all included individuals are depicted in Table 1.

### 3.2. PAT Volume and CAC

The determination of PAT by CT showed a high reliability. We determined a very low inter-observer variability of 3.3%.

The mean PAT volume was 215 ± 107 mL in all patients. The PAT volume in smokers was significantly higher compared to PAT volume in non-smokers, 231 ± 104 mL versus 201 ± 99 mL, *p* = 0.03, see Table 1.

We could observe a constant increase in PAT volume with a number of cardiovascular risk factors, up to 320 ± 157 mL in patients with five cardiovascular risk factors. As depicted in Figure 2, patients without cardiovascular risk factors showed a significantly lower PAT volume of 153 ± 155 mL compared to patients with more than one risk factor, *p* < 0.05.

### 3.3. PAT Volume and Risk Factors

In the multivariate analysis age, male sex, and BMI could be identified as independent factors for a PAT volume above 250 mL. In addition, the cardiovascular risk factors (hypertension, hyperlipidemia, diabetes mellitus, and smoking) could be identified as independent risk factors for a PAT volume above 250 mL after correction for the accompanying risk factors and age, sex, and BMI as depicted in Table 2. Smoking as a non-metabolic risk factor could be identified as an independent risk factor for elevated PAT with a OR 2.92 [2.31, 3.61], *p* < 0.001. We found a variance inflation factor < 2, indicating that a significant multicollinearity is not to be assumed.

## 4. Discussion

The aim of this study was to evaluate the extent of PAT in smokers compared to non-smokers. Different modalities for the quantification of PAT have been established in the recent years. Echocardiography has been used to determine the thickness of epicardial fat anteriorly to the right ventricle as a marker of the extent of PAT. The thickness correlated very well to the amount of visceral adipose tissue determined by MRI [29]. However, evaluation of PAT volume by CT scans showed most accurate values. In our study the assessment of PAT by CT showed a high reliability, which is shown by low inter-observer variability of 3.3%. Scans could be obtained in all patients with adequate quality. No patient was closed out due to technical reasons.

In our analysis, pericardial adipose tissue was defined as epicardial fat plus paracardial fat in contrast to some previous studies that defined PAT as paracardial fat alone. In CT studies without contrast agent enhancement, the differentiation between epicardial and pericardial fat is limited. In order to achieve a high diagnostic accuracy and reproducibility, we therefore defined PAT as paracardial and epicardial fat. As we could already demonstrate, in earlier studies, that there is an excellent correlation between epi- and pericardial fat [7,30]. Thus, PAT as paracardial and epicardial fat can be used to quantify pericardial fat.

We evaluated PAT as a promoter of atherosclerotic processes. Especially in patients with metabolic risk factors, PAT acts multifunctionally and induces inflammation, neovascularization, and oxidative stress to the coronary arteries by production and secretion of multiple endocrine and paracrine factors [12]. An increased volume of PAT can be observed in patients with diabetes mellitus, hyperlipidemia, and suspected metabolic syndrome [12,31]. PAT has a strong correlation to total visceral abdominal tissue but was independent from BMI [7,30]. In accordance with these findings and results in former CT studies we found an increased PAT volume in patients with diabetes mellitus hyperlipidemia.

Though PAT volume increased with CAC score, in all CAC score groups, a significant part of patients showed PAT volumes within the lowest and highest quartile. This indicates the wide range of PAT volumes, independent of CAC, and illustrates the PAT volume provide an additional information on the individual atherosclerotic activity. A similar distribution of PAT and CAC could be found in a prior study by our group, showing the additional value of PAT in the prediction of the PAT independent from CAC score [19].

Besides these metabolic risk factors, we wanted to evaluate the association of smoking as a non-metabolic risk factor on PAT. Cigarette smoke leads to endothelial dysfunction, enhanced thrombosis, and a combination of vascular and systemic inflammation [23,32,33]. Cigarette smoke induces a sustained endothelial-platelet and endothelial-leukocyte activation by expression of proinflammatory cytokines and adhesions molecules [26]. Smoking induces higher amounts of TNF-α and interleukin-6, which leads to an increase of inflammatory signals such as nuclear factor kappa B (NF-κB), Phosphoinositide 3-kinases, jun-N-terminal kinase cascades, and induces lipolysis in periadipocytes [34,35]. Nicotinic receptors in adipose tissue modulates inflammatory gene expression in human adipocytes [36]. However, because of the huge number of toxic chemicals in cigarette smoke the underlying mechanisms for atherosclerosis are not completely clarified.

Besides proinflammatory mechanisms, smoking mediates direct effects on adipose tissue. Polycyclic aromatic hydrocarbons (PAHs), another class of compounds in cigarette smoke, downregulate cholesterol efflux and induce atherogenesis [37]. Newer studies found that nicotine reduces phosphodiesterase activation to promote lipolysis resulting in an increase in circulating free fatty acids levels [38]. Smoking results in a significant reduction in adiponectin and leptin, which modulates insulin effects in adipose tissue [36]. Further, there is a significant inverse correlation by adiponectin and PAT [7]. As shown earlier by our group and others, hs-CRP was significantly associated with increased pericardial fat tissue [7,31]. These inflammatory signals were induced by smoking as well and are independent from BMI [7,21]. Considering these multiple effects, we assumed an increase in PAT volumes in comparison to non-smokers.

Actually, we found a significantly higher volume of PAT in smokers compared to non-smokers, independent from BMI, metabolic, or other cardiovascular risk factors. Increased PAT volume may reflect the increased inflammatory activity in smokers, which is responsible for the increased cardiovascular risk in smokers in comparison to non-smokers. As shown in a previous study, CAC alone underestimates the cardiovascular risk in smoker in comparison to non-smokers [4]. Additional determination of PAT might, therefore, be a useful tool to improve risk stratification in particular, as PAT volume can be assessed easily from data sets acquired for calcium screening. It has already been shown before that PAT combined with CAC might improve risk stratification for future cardiovascular events [20,39].

It is assumed that patients revealing only noncalcified plaques already reveal significantly elevated PAT volumes. In addition, elevated PAT volume was associated with the presence of vulnerable plaque components, independent of BMI [40]. This indicates that PAT volume accumulation may precede plaque calcification and the development of mature atherosclerotic plaques in general [7,19].

The strength of the study is the large population that had been evaluated during a cardiological preventive examination. Hereby, the PAT could be measured in all patients with a low inter-observer variability. We demonstrated that smokers had significantly more PAT than non-smoker in a large population, which supports findings that smoking leads to an increase in PAT in longitudinal studies [41]. Our observation was independent of metabolic risk factors such as diabetes, BMI, and hyperlipidemia. Several limitations of the analysis should be noted. A limitation is that our study is a retrospective non-randomized single center study of patients sent to our institution for a preventive medical check-up. Therefore, our study population cannot be considered as an unselected population. The study population consisted of a Caucasian population who was asymptomatic and had no former history of coronary events. However, due to distribution of risk factors and PAT, our study population reveals a typical population with cardiovascular risks in Europe. Another limitation of our study is that the extent of smoking was not quantified, as reliable data regarding amount of pack years was not available. We had applied the CDC definition of smoking, but the number of pack years was not registered. Thus, we could not examine a possible relation between the extent of smoking by pack years and PAT.

## 5. Conclusions

Determination of PAT as a marker of individual atherosclerotic activity and inflammatory burden offers additional information in cardiac risk prediction. Our data showed an increased PAT volume in smokers, independent from accompanying metabolic risk factors. Therefore, this imaging modality might illustrate the increased inflammatory activity in smokers in comparison to non-smokers. Confirmation of the study findings should be performed in larger populations, and longitudinal studies should be performed.

## Figures and Tables

**Figure 1 jcm-10-03382-f001:**
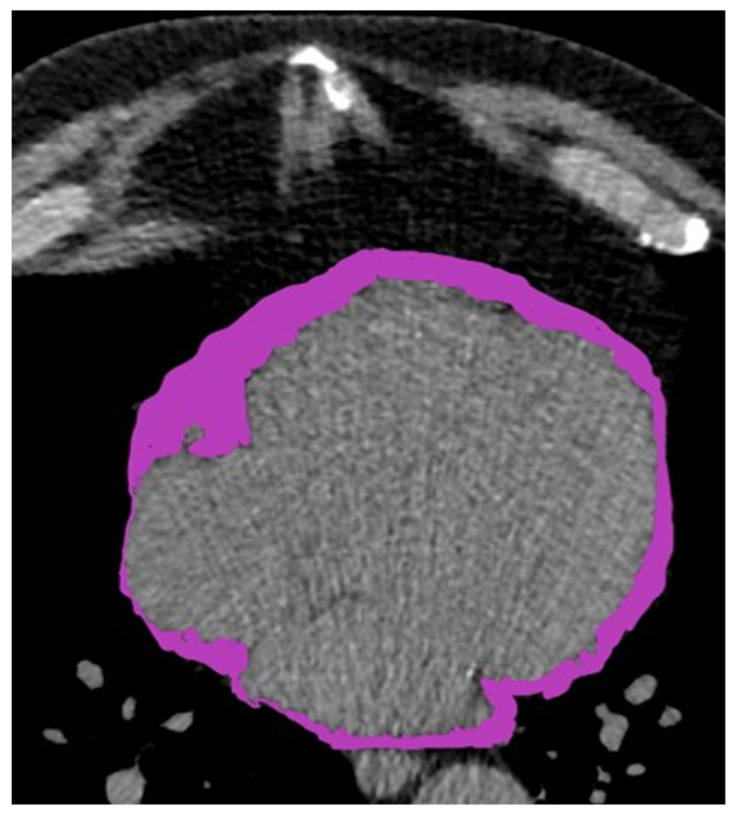
Extended pericardial adipose tissue in a smoker. PAT volume was 253 mL.

**Figure 2 jcm-10-03382-f002:**
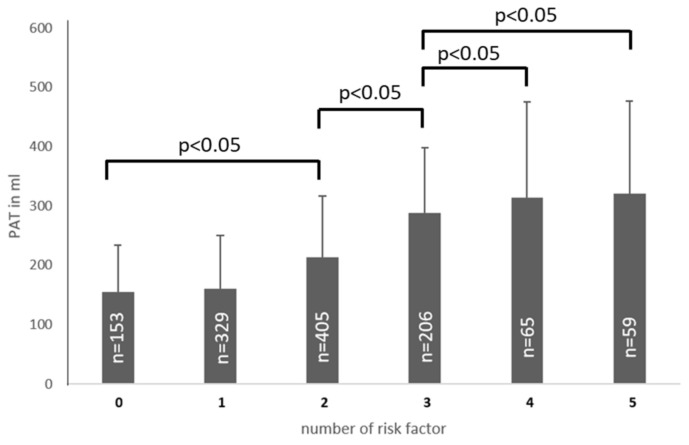
Number of risk factors increases PAT (in mL). A significant higher PAT volume could be measured in patients with 2 risk factors, in patients with 3 risk factors vs. 0–2 risk factors, in patients with 4 vs. 0–3 risk factors an in patients with 5 risk factors vs. 0–3 risk factors. The mean CAC score was 287 ± 188 (range 0–1208). The mean CAC score in smokers was 305 ± 191, compared to 269 ± 191 in non-smokers (*p* < 0.01). PAT distribution in the different CAC groups is depicted in Figure 3. PAT volume increased with CAC score with a wide range of PAT volumes in each CAC score groups. In the low CAC score group (CAC score 0–10) 16.9% possessed a PAT volume within the 4th quartile, whereas 17% in the high CAC group (>400) showed a PAT volume within the 1st quartile.

**Figure 3 jcm-10-03382-f003:**
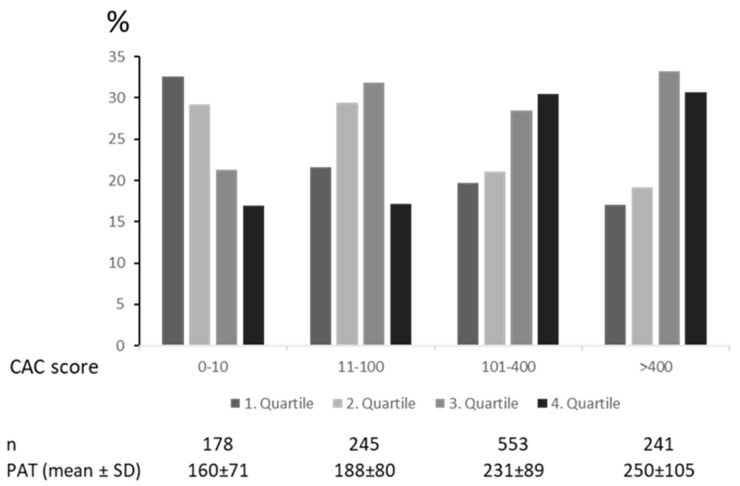
Distribution of PAT quartiles in different CAC groups.

**Table 1 jcm-10-03382-t001:** Baseline characteristics of 1217 patient included in the study.

	All Patients		Non-Smokers		Smokers		*p*-Value
	*n*	%	*n*	%	*n*	%	
patients	1217		644	52.9	573	47.1	
male	727	59.7	374	30.7	353	29	0.24
female	490	40.3	270	22.2	220	18.1	0.29
age	58.3 ± 8.3		57 ± 7.9		59.9 ± 8		0.41
BMI	27.2 ± 4.8		27 ± 4.5		27.5 ± 4.9		0.28
BMI > 30 kg/m^2^	389	32.0	215	33.4	174	30.5	0.27
arterial hypertension	602	49.5	312	25.6	290	23.8	0.24
hyperlipidemia	465	38.2	245	20.1	220	18.1	0.31
diabetes	171	14.1	91	7.5	80	6.6	0.29
family history of CAD	584	48	301	24.7	283	23.3	0.15
average number of risk factors	1.9		1.4		2.5		<0.001
PAT	215 ± 107		201 ± 99		231 ± 104		0.03

CAD: coronary artery disease, PAT: pericardial adipose tissue.

**Table 2 jcm-10-03382-t002:** Odds ratio of different risk factors adjusted for age, sex, hyperlipidemia, diabetes, arterial hypertension, and smoking for a PAT above 250 mL in a multivariate analysis.

	OR [95% CI]	*p*-Value
age	1.10 [1.06, 1.14]	<0.001
BMI	1.19 [1.09, 1.32]	<0.001
male sex	1.20 [1.15, 1.30]	<0.001
hypertension	1.80 [1.60, 2.04]	<0.001
hyperlipidemia	2.84 [2.31, 3.39]	<0.001
diabetes	2.31 [2.04, 2.61]	<0.001
smoking	2.92 [2.31, 3.61]	<0.001

## Data Availability

The data presented in this study are available on request from the corresponding author for researchers who meet the criteria for access to confidential data. The data are not publicly available. The data cannot shared publicly because of the privacy of individuals participated in this study. Data are available from the Ethics Committee (contact via gregor.zimmermann@tum.de).

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
