# Peer review of "Increased Pericardial Adipose Tissue in Smokers"

_jcm, 2021, doi:10.3390/jcm10153382_

Round 1
Reviewer 1 Report
I read the article entitled "Increased pericardial adipose tissue in smokers". Here the authors report findings from a large cohort of patients imaged by CT, in which pericardial adipose tissue was assessed. They found increased adipose tissue in smokers vs non-smokers. Furthermore, the association between smoking habit and increased adipose tissue remained significant after correction in multivariable analysis. The authors concluded that this finding could represent the increased inflammatory activity in smokers vs non-smokers. The study design and methodology of data collection are accurate and scientifically sound. I have the following comments:
1) My main concern is related to authors' conclusions that might include overinterpretation of study findings. Specifically, despite being significant, the absolute difference in adipose tissue between smokers vs non-smokers was relatively small. Furthermore, it is not clear the independent role of smoking habit as compared to other CV risk factors, which all resulted significant multivariable analysis (multicollinearity should be assessed). Perhaps, inclusion of inflammaotry biomarkers assessment and their inter-relationship with smoking habit and pericardial adipose tissue extent could help making the study more robust.
2) I would avoid the term "impact", as a cause-effect relationship cannot be established by study design
3) Do authors have any data regarding coronary artery calcium score? Can any cluster be identified (i.e. smoke habit+increased PAT+low calcification vs patients with extensive coronary calcifications)?
Author Response
Submitting Original Article to Journal of Clinical Medicine
Title: “Increased pericardial adipose tissue in smokers”
Dear Editor,
Thank you very much for receiving our manuscript and considering it for review. Enclosed you will find the revised version of our manuscript, which we would like to resubmit to Journal of Clinical Medicine.
We have addressed each of the journal requirements, the editors ‘comments and the review comments to the author. By responding to the detailed and thoughtful comments of the reviewer, the paper has been substantially improved. Changes in the manuscript are marked.
Thank you very much for reconsidering our manuscript for publication in Journal of Clinical Medicine. We appreciate your time and look forward to your response.
Sincerely,
Gregor Zimmermann, M.D.
Department of Internal Medicine I
Klinikum rechts der Isar
Technical University Munich
Ismaninger Str. 22
81675 Munich, Germany
Tel.: +49 89 41405803, Fax.: +49 89 4140
Reviewer Comments:
Reviewer 1:
- My main concern is related to authors' conclusions that might include overinterpretation of study findings. Specifically, despite being significant, the absolute difference in adipose tissue between smokers vs non-smokers was relatively small. Furthermore, it is not clear the independent role of smoking habit as compared to other CV risk factors, which all resulted significant multivariable analysis (multicollinearity should be assessed). Perhaps, inclusion of inflammatory biomarkers assessment and their inter-relationship with smoking habit and pericardial adipose tissue extent could help making the study more robust.
Thank you for this helpful advice. To assess multicollinearity in our population we calculated variance inflation factor (VIF) for the different cardiovascular risk factors. We included this in the methods section of the revised manuscripts.
For all risk factors we could find a VIF below 2, indicating that a significant multicollinearity is not to be assumed. We added this information in the results section.
We agree with the reviewer's recommendation that an analysis of inflammatory markers would be helpful to analyze the inter-relationship between smoking status and PAT. In particular, an analysis of hs-CRP would further support this theory in our collective. However, we could not examine these data in our population because these biomarkers were not collected from a large portion of our population and therefore were not included in this study. In previous collectives of our research group, we demonstrated that increased PAT is significantly associated with increased levels of TNF-alpha and hs-CRP (Pericardial adipose tissue determined by dual source CT is a risk factor for coronary atherosclerosis, Greif M et al, DOI: 10.1161/ATVBAHA.108.180653). We added this information in the manuscript.
- I would avoid the term "impact", as a cause-effect relationship cannot be established by study design.
We appreciate this comment and corrected this in the revised manuscript.
- Do authors have any data regarding coronary artery calcium score? Can any cluster be identified (i.e. smoke habit+increased PAT+low calcification vs patients with extensive coronary calcifications)?
Thank you for this helpful advice. We added the CAC score of our study population in the revised manuscript. As suggested, we created different CAC groups and analyzed the PAT distribution in these groups. We added Figure 3 to provide more information regarding CAC in our population.
Though we could find an increase of PAT in higher CAC score groups, a wide range of PAT volumes could be detected in each CAC score groups, indicating that PAT volume provides an individual information of atherosclerotic activity.
Figure 3. Distribution of PAT quartiles in different CAC groups.

Reviewer 2 Report
- You could maybe mention previous studies focused on smoke and PAT. By example PMID 30047503
- Why not showing results of calcium scores in the population? It would be of interest to provide also an analyze with correlation between CS and smoking status in your population. Especially because these CT scan have been performed for this assessment.
Author Response
Submitting Original Article to Journal of Clinical Medicine
Title: “Increased pericardial adipose tissue in smokers”
Dear Editor,
Thank you very much for receiving our manuscript and considering it for review. Enclosed you will find the revised version of our manuscript, which we would like to resubmit to Journal of Clinical Medicine.
We have addressed each of the journal requirements, the editors ‘comments and the review comments to the author. By responding to the detailed and thoughtful comments of the reviewer, the paper has been substantially improved. Changes in the manuscript are marked.
Thank you very much for reconsidering our manuscript for publication in Journal of Clinical Medicine. We appreciate your time and look forward to your response.
Sincerely,
Gregor Zimmermann, M.D.
Department of Internal Medicine I
Klinikum rechts der Isar
Technical University Munich
Ismaninger Str. 22
81675 Munich, Germany
Tel.: +49 89 41405803, Fax.: +49 89 4140
Reviewer Comments:
Reviewer 2:
- You could maybe mention previous studies focused on smoke and PAT. By example PMID 30047503.
Thank you for this helpful advice. We added this information in revised manuscript and in the references.
- Why not showing results of calcium scores in the population? It would be of interest to provide also an analyze with correlation between CS and smoking status in your population. Especially because these CT scan have been performed for this assessment.
The reviewer raised an interesting question. We added CAC score for the study population, smokers and non-smokers. In addition, we added the PAT volume in different CAC groups as depicted in Figure 3 in the result section of the revised manuscript.

Round 2
Reviewer 1 Report
Thank you for addressing all the issues. I have no further comments